# The Contribution of the Brain–Gut Axis to the Human Reward System

**DOI:** 10.3390/biomedicines12081861

**Published:** 2024-08-15

**Authors:** Katerina Karaivazoglou, Ioanna Aggeletopoulou, Christos Triantos

**Affiliations:** 1Department of Psychiatry, University Hospital of Patras, 26504 Patras, Greece; karaivaz@hotmail.com; 2Division of Gastroenterology, Department of Internal Medicine, University Hospital of Patras, 26504 Patras, Greece; chtriantos@upatras.gr

**Keywords:** brain–gut axis, reward network, microbiota, enteric neuropeptides

## Abstract

The human reward network consists of interconnected brain regions that process stimuli associated with satisfaction and modulate pleasure-seeking behaviors. Impairments in reward processing have been implicated in several medical and psychiatric conditions, and there is a growing interest in disentangling the underlying pathophysiological mechanisms. The brain–gut axis plays a regulatory role in several higher-order neurophysiological pathways, including reward processing. In this context, the aim of the current review was to critically appraise research findings on the contribution of the brain–gut axis to the human reward system. Enteric neuropeptides, which are implicated in the regulation of hunger and satiety, such as ghrelin, PYY_3–36_, and glucagon-like peptide 1 (GLP-1), have been associated with the processing of food-related, alcohol-related, and other non-food-related rewards, maintaining a delicate balance between the body’s homeostatic and hedonic needs. Furthermore, intestinal microbiota and their metabolites have been linked to differences in the architecture and activation of brain reward areas in obese patients and patients with attention deficit and hyperactivity disorder. Likewise, bariatric surgery reduces hedonic eating by altering the composition of gut microbiota. Although existing findings need further corroboration, they provide valuable information on the pathophysiology of reward-processing impairments and delineate a novel framework for potential therapeutic interventions.

## 1. Introduction

Almost 2500 years ago, Hippocrates insightfully suggested that the origins of every disease lie in the gut, emphasizing the importance of the gastrointestinal system for body homeostasis and human health. Research has repeatedly shown that the gastrointestinal tract, apart from its obvious contribution to food intake and digestion, constitutes the body’s critical hub where complex interactions take place between the internal and the external milieu and between internal bodily systems, including the immune, the peripheral, and the central nervous system [1,2]. These interactions are mediated by a perplexing network of signaling pathways that encompass the gut microbiota, the intestinal wall, the enteric nervous system, the vagus nerve, the sympathetic ganglia, and several cortical and subcortical regions and constitute the microbiota–gut–brain axis [1]. A major role of this bidirectional communication is to inform the brain about internal and external changes and guide appropriate physiological responses in order to maintain energy balance, preserve one’s health status, and ultimately favor adaptive behavior and survival. In this respect, the gut–brain axis regulates neural and inflammatory signaling throughout the body and contributes to the architecture and functioning of key brain networks subserving a variety of functions ranging from simple reflexes to higher-order cognitive processing [2].

The human brain reward system is a set of functionally interconnected neural circuits that process stimuli which are associated with desire, pleasure, behavioral responses, and associative learning [3,4]. It consists of mesolimbic pathways, including the ventral tegmental area, the nucleus accumbens, the amygdala, the striatum, and the septum, and of meso-corticolimbic pathways, involving the hippocampus and the prefrontal cortex and the parahippocampal, entorhinal, and motor function-related cortical areas. There is a wealth of animal research linking the brain–gut axis to the shaping and function of the reward system. In addition, there is a growing number of studies focusing on the role of the gut–brain communication on the human reward system [3,4]. These studies are characterized by remarkable heterogeneity. Firstly, they include a wide range of participants who are either healthy volunteers, obese individuals, or patients with a variety of conditions which are associated with aberrant reward processing, such as drug and alcohol addictions, eating disorders, and major depression. In addition, they assess the role of distinct parts of the brain–gut axis, including microbiota and their metabolites, gut peptides and hormones, and the vagus nerve. For this reason, relevant findings need to be critically appraised, categorized, and integrated in a comprehensive and concise manner to inform scientific knowledge and clinical practice.

In this context, the aim of the current review is to descry and scrutinize original articles focusing on the associations between the brain–gut axis and the human reward system. We conducted a rigorous search of PubMed and ScienceDirect using all possible combinations of relevant terms to extract as much information as possible (Table 1).

We exclusively included studies conducted on human subjects, which were published during the last decade (2014–2024) and were written in English. Furthermore, we manually searched all references from the selected articles for additional information. The final search was conducted at the beginning of June 2024. Table 2 includes all studies which were considered relevant and whose findings were reviewed in the current article.

## 2. Characteristics of Studies

Our search retrieved 22 original research articles which were considered relevant. A total of 7 studies included healthy subjects [5,6,16,17,19,20,25]; 7 studies focused on obese or overweight subjects [7,10,11,12,13,24,26]; 4 studies included patients with alcohol-use disorder (AUD) [8,15,22,23]; 2 studies included patients with anorexia nervosa [9,21]; 1 study focused on patients with major depressive disorder [14]; and 1 study included patients with attention deficit and hyperactivity disorder (ADHD) [18]. Our search did not reveal any relevant article focusing on any other psychiatric disorder, except for the aforementioned mental health conditions. A total of 4 investigations included only female participants [9,11,12,14]; 1 study included only male participants [20]; and the remaining studies included both genders [5,6,7,8,10,13,15,16,17,18,19,21,22,23,24,25,26]. Although all studies originated from Western countries and most participants were of Caucasian White origin, a majority of the samples also included individuals with a variety of racial affiliations, such as Hispanic, African American, and Asian. Moreover, a vast majority of the studies exclusively included adults, except for 1 investigation that recruited both adolescents and young adults [9], while no study included participants over 70 years of age. A total of 13 studies had a cross-sectional design [5,6,7,9,10,11,13,14,17,18,22,24,25]; 4 trials were prospective [12,16,19,21]; and 5 were randomized controlled trials [8,15,20,23,26]. A vast majority of the reviewed studies included a neuroimaging technique in their methodology; 19 studies used brain MRIs [5,6,7,8,9,10,11,12,13,14,15,16,17,18,20,22,24,25,26]; and 1 study used H_2_^15^O PET [19]. The composition of gut microbiota was determined with the use of 16s ribosomal RNA gene sequencing and gut neuropeptides, and microbiota metabolite levels were measured with the use of immunoassay methods.

## 3. Healthy Subjects

Earlier prospective neuroimaging studies in healthy volunteers [16,19,25] have shown that gut-derived hormone levels such as ghrelin and peptide PYY_3–36_ are involved in the activation of brain areas involved in the hedonic control of food intake. More specifically, ghrelin, which is a stomach-derived hormone and acts as the main orexigenic hormone to induce hunger and feeding, affects activation patterns and dopaminergic transmission in the ventral striatum and the ventromedial prefrontal cortex. In addition, it strengthens the functional connectivity between the hippocampus and the ventral striatum, increasing the incentive value of food-cued stimuli, thus enhancing associative learning with food rewards [16]. In a similar way, decreased ghrelin and increased PYY_3–36_ levels inhibit dopaminergic transmission in the ventral tegmental area, leading to a decline in the reward value of food, thus signaling meal termination [19,25]. There is also research on the involvement of ghrelin in the processing of other non-food rewards, including social rewards; however, relevant findings are conflicting. Sailer et al. [6] revealed that ghrelin levels are associated with one’s response to non-social rewards and the related activation of the ventromedial prefrontal cortex but are not associated with the activation of the ventral striatum and the subsequent response to social rewards. In contrast, a recent fMRI study revealed that ghrelin may reduce the incentive value of social rewards, given that decreased post-prandial ghrelin levels increase the activation of the right medial orbitofrontal cortex, which is associated with the increased pleasantness of a social cue (touch) [5]. Colonic propionate, which is a short-chain fatty acid (SCFA) produced by the microbiota, induces a decreased anticipatory reward response to high-energy foods, as was evident by the reduced activation of the caudate and the nucleus accumbens in the ventral striatum in a small sample of healthy males [20]. Other microbiota-derived fecal metabolites, including indole, indolacetic acid, and scatole, have also been correlated with functional and structural connectivity patterns between the amygdala and the nucleus accumbens and the amygdala and the anterior insula. These networks are involved in reward processing and affect hedonic food intake both in obese and normal-weighted individuals [17]. The authors of this study suggest that the effects of microbiota-derived metabolites on brain reward regions may be mediated by enteroendocrine L-cells and the production of glucagon-like peptide 1 (GLP-1) or by the regulation of serotonin synthesis and release.

## 4. Obesity

Obese individuals commonly report disturbed eating behaviors, impaired self-control, and the aberrant activation of reward-processing brain regions [27], and there are accumulating findings linking these disturbances with the gut–brain axis. In a recent series of MRI studies, obese individuals presented with structural and functional alterations in key areas of the brain reward network, and these alterations were correlated with the composition of microbiota and fecal amino acid levels [7,11,13]. More specifically, Hung et al. [13] showed that in obese subjects, the structural volume of the left hippocampus is positively correlated with Bacteroidetes, Firmicutes, and Actinobacteria; the structural volume of the right amygdala is positively correlated with Actinobacteria; the structural volume of the nucleus accumbens is positively correlated with Fusobacteria; and the structural volume of the brainstem is negatively correlated with Proteobacteria. Furthermore, the structural volumes of several reward-related brain regions are negatively associated with fecal amino acid levels [13]. Likewise, a large well-designed fRMI study revealed that the *Prevotella*/*Bacteroides* ratio was independently correlated with the central connectivity pattern of the left nucleus accumbens in a sample of obese and normal-weighted subjects [7]. In addition, specific microbiota taxonomic profiles, including the genera *Megamonas*, *Bacteroides*, *Eubacterium*, and *Akkermansia*, were associated with the anatomical connectivity between the putamen and the brainstem and the putamen and the intraparietal sulcus/transverse parietal sulcus in a sample of female subjects encompassing obese, overweight, and normal-weighted individuals [11]. However, in both of these studies, the aforementioned associations between the composition of microbiota and the brain reward system were observed in the whole sample, and no separate analysis was performed between obese and non-obese individuals [7,12].

In another earlier fMRI randomized controlled study, obese patients who had undergone a Roux-en-Y gastric bypass (RYGB) displayed attenuated brain-hedonic responses to food and reported a lower palatability and a lower appeal for high-caloric food compared to patients who had undergone gastric banding surgery and to patients who were unoperated. These differences in the activation of the reward system (orbitofrontal cortex, amygdala, anterior insula, nucleus accumbens, and caudate) were mediated by increased plasma levels of PYY_3–36_ and GLP-1 [24]. Likewise, a laparoscopic sleeve gastrectomy (LSG) was found to lead to the reduced connectivity between the putamen and the precuneus, which are brain areas associated with hedonic eating, through changes in the composition of gut microbiota and their metabolites, namely, an increase in *Bacteroides*, *Ruminococcus*, and *Holdemanella* and a decrease in 1-palmitoyl-2-palmitoleoyl levels [12]. In another investigation, decreased ghrelin levels were associated with the increased reward-related activation of the dorsolateral prefrontal cortex and the precuneus/posterior cingulate in obese individuals compared to controls [10]. It should be noted that in that study, ghrelin levels were associated with non-food reward processing, suggesting that there are probably distinct neurophysiological pathways mediating hedonic responses to different types of rewards. Another gut-derived peptide which is involved in the regulation of food intake is GLP-1 [10]. Obese patients with diabetes mellitus type 2 and obese normoglycemic subjects displayed attenuated brain responses to food pictures in the appetite and reward-related areas after the IV administration of a GLP-1 receptor agonist [26].

## 5. Psychiatric Disorders

Abnormalities in reward processing characterize a variety of psychopathological conditions, including addictions, mood disorders, impulse control disorders, eating disorders, and attention deficit and hyperactivity disorder (ADHD) [28,29]. According to the current literature search, relevant studies on psychiatric populations exclusively involved patients with alcohol-use disorder, anorexia nervosa, major depression, and ADHD.

### 5.1. Alcohol-Use Disorder

There is a growing interest in disentangling the contribution of reward processing in generating and maintaining psychopathological symptoms, and, recently, a constantly rising number of studies have focused on the role of several gut–brain peptides in modulating reward processing and shaping addictive behaviors. Most of these findings originate from preclinical studies; however, there is a small but significant number of relevant research on human subjects too [30,31]. An earlier human genetic study revealed that certain polymorphisms of the GLP-1 receptor gene are associated with reward-related behaviors, including alcohol abuse and smoking and the greater reward-related activation of the right globus pallidus in patients with alcohol-use disorder [22]. In addition, a recent randomized placebo-controlled fMRI study revealed that the administration of a GLP-1 receptor agonist in a group of patients with severe alcohol-abuse disorder, exenatide; although, it did not significantly decrease heavy drinking, it did attenuate alcohol-related-cue reactivity in the ventral striatum and the septal area, which are implicated in reward processing and addiction [8]. Similarly, in two randomized, double-blind, placebo-controlled studies, the intravenous administration of ghrelin increased alcohol seeking in heavy drinkers [15,23] and induced the alcohol-related activation of the amygdala, which constitutes a crucial hub of the reward network [15].

### 5.2. Attention Deficit Hyperactivity Disorder and Depression

Research has shown that several components of the gut–brain axis, including intestinal microbiota and their metabolites, the vagus nerve, and neuropeptides, have been implicated in the pathophysiology of a variety of mental disorders, including depression and attention deficit hyperactivity disorder (ADHD) [32,33]. In addition, both major depressive disorder and ADHD are commonly characterized by a low hedonic tone that is a genetically determined decreased capacity to experience pleasure which is mediated mainly by the hypo-activation of the mesolimbic dopaminergic system and a wider under-responsiveness of the reward-processing network [34]. In this context, there is a very limited yet important line of findings focusing on the role of the gut–brain axis in modulating reward processing in depressive and ADHD patients. In a recent small-scale study including female patients with major depression, post-prandial ghrelin levels were elevated in hypophagic patients compared to hyperphagic patients and healthy controls. In addition, ghrelin levels were positively associated with the response of the ventral tegmental area bilaterally and the left hypothalamus to high-calorie food in hyperphagic patients and were negatively associated with the response of the right hypothalamus to food cues in hypophagic patients [14]. Likewise, Aarts et al. [18] found microbiota taxonomic differences between adolescents and young adults with ADHD compared to healthy controls, with an abundance of Actinobacteria in the ADHD patients and an increased capacity of the gut microbiome of the ADHD patients to produce monoamine precursors (phenylalanine). In addition, the ADHD patients exhibited a decreased ventral striatum activation for reward anticipation compared to the control group, while the abundance of microbiota cyclohexadienyl dehydratase (CDT) genes involved in phenylalanine production negatively correlated with reward anticipation responses in the ventral striatum bilaterally.

### 5.3. Eating Disorders

Eating disorders are a group of disabling, costly, and, in some cases, life-threatening mental disorders characterized by abnormal attitudes towards one’s weight, body image, and eating [35]. There is a wealth of data revealing impaired neurocognitive and behavioral patterns of reward processing in eating disorders, mostly anorexia nervosa and bulimia nervosa [36,37]. More specifically, anorexia nervosa has been associated with hyporesponsiveness to reward and abnormal reward attribution, while bulimia nervosa has been linked to an increased expectancy of rewards but a decreased experienced food-related reward [37]. Several neuroimaging studies have shown significant structural and functional alterations of the brain circuitry subserving the anticipation and processing of rewards in patients with eating disorders. Furthermore, there are emerging data suggesting that these alterations may be associated with the effects of several neuropeptides, such as ghrelin and PYY_3–36_, released by the L-cells of the intestinal wall [36].

Animal studies and, to a lesser extent, human trials have repeatedly shown that ghrelin modulates not only homeostatic food intake but also hedonic eating and other reward-related responses through its action on the mesocorticolimbic dopaminergic system [9]. An earlier small-scale study showed that in symptomatic anorexia nervosa patients, there was a disruption in the ghrelin response to hedonic eating compared to weight-restored patients and healthy controls, indicating that underweight anorexia nervosa patients experience diminished food-related anticipatory and consummatory rewarding feelings [21]. Likewise, a more recent wide-scale investigation revealed that anorexia nervosa patients had increased ghrelin levels compared to healthy controls [9]. According to that study, symptomatic anorexia patients exhibited an increased capacity to delay gratification and resist immediate rewards compared to healthy controls with similar ghrelin levels [9], probably suggesting an altered sensitivity of the brain-reward areas to ghrelin signaling in anorexia nervosa.

## 6. Discussion

The current paper detected and critically reviewed original articles focusing on the role of the brain–gut axis in modulating the human reward system. Most studies were performed on healthy volunteers and obese individuals, but there were also a small number of investigations that included patients with alcohol-use disorder, eating disorders, major depression, and ADHD (Figure 1). In general, the studies’ samples were, to some extent, representative of both genders and of various races; however, in terms of age, children and elderly individuals were almost completely missing.

Relevant findings reveal that gut-derived hormones and intestinal microbiota and their metabolites are associated with structural and functional alterations of the reward network. Several studies have shown that ghrelin is associated not only with homeostatic eating but also with hedonic eating and other non-food-related rewards in healthy individuals and obese patients and patients with alcohol-use disorder, major depression, and anorexia nervosa [5,6,9,10,15,16,21,23]. Ghrelin levels have been repeatedly linked to activation patterns of certain areas of the reward-processing network, including dopaminergic transmissions in the ventral tegmental area and the ventral striatum, the ventromedial prefrontal cortex, and the amygdala [16]. More specifically, ghrelin seems to increase brain responses to food- and non-food-related rewards and may attenuate the brain’s response to social rewards [5,6,16]. However, in anorexia nervosa patients, there seems to be a disrupted response of the brain reward network to ghrelin secretion, which might explain why these patients experience less food-derived feelings of pleasure and display an exaggerated tendency to delay gratification and resist immediate rewards [9,21]. In contrast to ghrelin’s effects, PYY_3–36_ and GLP-1 have been associated with the decreased activation of brain reward areas in response to food- and alcohol-related cues in heathy controls and obese and diabetic patients and patients with alcohol-use disorder [19,24,25,26]. In total, these findings suggest that enteric hormones modulate the way the brain attributes incentive values to a variety of different stimuli, thus shaping pleasure-seeking behaviors. In this way, they seem to regulate a delicate balance between the body’s metabolic and homeostatic needs on the one hand and the natural human tendency towards intrinsic, reward-related motivations on the other hand.

The composition of intestinal microbiota was significantly associated with the structural characteristics and connectivity patterns of several reward areas in normal-weighted and obese individuals and in patients with ADHD, suggesting that different microbiota genera are implicated in brain reward processing, affecting hedonic eating and other pleasure-seeking behaviors [7,11,13,18]. These effects are probably mediated by the release of fecal amino acids and other microbiota-derived metabolites which have been associated with distinct activation patterns in the reward circuitry, including connections between the amygdala, the nucleus accumbens, the insula, and the striatum [13,17,20]. In addition, bariatric surgery in morbidly obese individuals has been shown to inhibit hedonic eating through changes in the composition of gut microbiota and through the release of GLP-1 and PYY_3–36_ [24,26].

## 7. Conclusions and Future Directions

The current review met its purpose to present, in a coherent and systematic manner, all evidence regarding the role of the brain–gut axis in the modulation of the human reward system. To our knowledge, this is the first review that focuses on this specific topic, including research originating from a wide range of clinical conditions and categorized findings by participants’ clinical status. In this way, we maximized the clinical usefulness of our observations. Furthermore, although our search revealed that the relevant literature is rather limited, it yielded significant novel information which emphasizes the key contribution of gut–brain interactions to the shaping of human behaviors. Most studies included in this review had a solid methodology, such as neuroimaging, 16s ribosomal RNA gene sequencing, and immunoassays and have provided important evidence that human pleasure-seeking behaviors are partly determined by low-level pathophysiological processes localized within the gastrointestinal tract. Several components of the brain–gut axis, including microbiota, microbiota-derived metabolites, enteric neuropeptides, and neurotransmitters, either enhance or inhibit reward-related signaling within the brain, thus affecting the formation of habits, other goal-directed behaviors, and their accompanying feelings.

In total, these findings suggest that the key to modifying dysfunctional behavioral patterns, for example, disordered eating, impulsive decision-making, or substance abuse, may lie, in part, on the therapeutic manipulation of the brain–gut axis. However, all this evidence needs to be corroborated and expanded on through larger-scale investigations encompassing a wider range of patient populations. In addition, more trials are needed to assess the effects of microbiota-modulating and enteric neuropeptide-based interventions on hedonic eating, addictive behaviors, risk-taking and impulsive behaviors, and other reward-related behaviors which constitute psychopathological symptoms and impair patients’ quality of life. Further exploring the pathophysiological connections between the brain–gut axis and the structure and function of the reward network might provide new therapeutic targets for a variety of medical and psychiatric conditions in order to more effectively manage patients’ symptomatology and alleviate their discomfort.

## Figures and Tables

**Figure 1 biomedicines-12-01861-f001:**
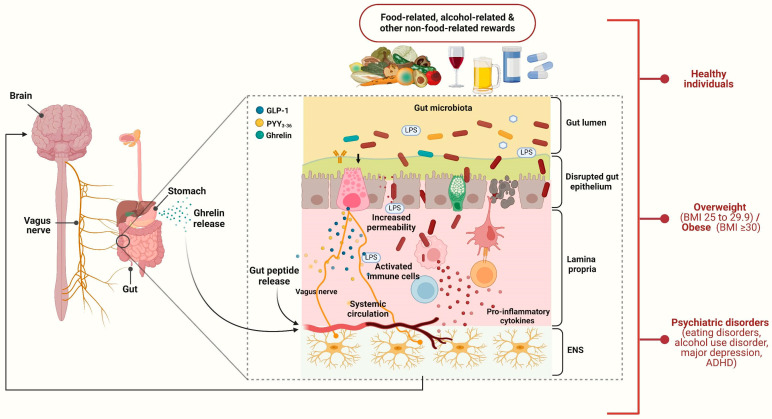
The role of the brain–gut axis in food-related and non-food-related rewards. This figure illustrates the interaction between the brain–gut axis and the reward system, highlighting how different physiological and pathological conditions influence this relationship. This complex interaction is described in the case of healthy subjects and obese or overweight individuals and patients with psychiatric disorders. These disorders include eating disorders, alcohol-use disorder, major depression, and attention deficit hyperactivity disorder (ADHD). This figure was created with BioRender.com. Abbreviations: PYY_3–36_, peptide YY_3–36_; GLP-1, glucagon-like peptide 1; LPS, lipopolysaccharide; ENS, enteric nervous system.

**Table 1 biomedicines-12-01861-t001:** Search terms used in this review.

Brain–gut axis synonyms and related terms	Reward system synonyms and related terms
Brain–gut axis, intestinal microbiota, microbiota metabolites, gut-derived hormones, ghrelin, glucagon-like peptide 1, PYY_3–36_	Reward system, reward processing, reward network, pleasure-seeking behaviors

**Table 2 biomedicines-12-01861-t002:** Original studies included in this review on the role of the brain–gut axis in the human reward system.

Author	Ref.	Country	Publication Year	Demographic and Clinical Characteristics of Participants	Design	Methodology
Pfabigan et al.	[5]	Norway and Sweden	2024	A total of 68 healthy volunteers (47 males); race: 66 of European descent and 2 of Asian descent; age: 18–55 yrs	CS	ELISA and brain fMRI during a CT-targeted touch task
Sailer et al.	[6]	Norway, Germany, and the UK	2023	A total of 68 healthy volunteers (47 males); race: 66 of European descent and 2 of Asian descent; age: 18–55 yrs	CS	ELISA and brain fMRI during a social-recognition-by-experts task and a social affirmation task
Dong et al.	[7]	The USA	2022	A total of 81 obese and 216 normal or overweight individuals; gender: 99 males; race: 110 Non-Hispanic White, 92 Hispanic, 65 Asian, 24 Black, and 6 Native American; age: 21–41.5 yrs	CS	16s RNA gene sequencing, metabolite analysis, and brain structural MRI and fMRI
Klausen et al.	[8]	Denmark, the USA, and Germany	2022	A total of 127 patients with AUD (76 males), with 62 randomized to exenatide and 65 randomized to a placebo; race: all White; mean age: 52 yrs	RCT	Symptom questionnaires, brain fMRI, and SPECT DAT scan
Bernardoni et al.	[9]	Germany	2020	A total of 94 acutely underweight female AN patients, 37 recovered female AN patients, and 119 female healthy controls; mean age: 16.1–22.2 yrs	CS	ELISA, brain MRI, delay discounting task, risk aversion for probabilistic gains (PDGs), and probabilistic losses tests
Bogdanov et al.	[10]	France, the USA, and Italy	2020	A total of 15 severely obese subjects (3 males) and 15 non-obese healthy controls (2 males); mean age: 37–38.7 yrs	CS	ELISA and brain fMRI during a guessing task
Dong et al.	[11]	The USA	2020	A total of 86 obese females with FA and 19 obese females without FA; race: 41 Hispanic, 28 Caucasian, 13 African American, 21 Asian, and 2 Other; age: 18–50 yrs	CS	16s ribosomal RNA gene sequencing, metabolomics, and brain MRI
Dong et al.	[12]	The USA	2020	A total of 18 female obese patients who underwent LSG; race: 8 Non-Hispanic White, 2 African American, 2 Asian, and 6 Hispanic; age: 18–55 yrs	Prospective trial	16s ribosomal RNA gene sequencing, mass spectrometry, and brain structural and rs-fMRI
Hung et al.	[13]	The USA	2020	A total of 130 overweight or obese individuals (43 males); race: 52 Hispanic and 78 Non-Hispanic; age: 18–60 yrs	CS	16s ribosomal RNA sequencing, mass spectroscopy, and brain structural MRI
Cerit et al.	[14]	The USA	2019	A total of 10 female hyperphagic MDD patients in remission, 18 female hypophagic MDD patients in remission, and 18 healthy controls; age: 22–42 yrs	CS	Brain fMRI during exposure to food pictures and radioimmunoassay
Farokhnia et al.	[15]	The USA	2018	A total of 11 heavy drinkers (8 males); race: 9 African American; mean age: 40 yrs	Double-blind RCT	Brain fMRI
Han et al.	[16]	Canada	2018	A total of 38 healthy subjects (21 males), who were administered either IV ghrelin or saline; mean age: 22.5 yrs	Single-blind, counterbalanced prospective trial	Brain fMRI during a food odor-conditioning task
Osadchiy et al.	[17]	The USA	2018	A total of 63 healthy individuals (29 males); age: 18–60 yrs	CS	Mass spectrometry, brain structural MRI, brain functional MRI, and diffusion-weighted MRI
Aarts et al.	[18]	The Netherlands	2017	A total of 19 ADHD patients (13 males) and 77 controls (41 males); mean age: 19.1–27.5 yrs	CS	16s ribosomal RNA gene sequencing fMRI
Ly et al.	[19]	Belgium	2017	A total of 15 healthy volunteers (8 males); mean age: 27.3 yrs	Prospective counterbalanced trial	RIA and H_2_^15^O-PET after a balloon- and nutrient-induced distension
Byrne et al.	[20]	The UK	2016	A total of 20 healthy non-obese men; race: 18 European Caucasian; age: 18–65 yrs	RCT	fMRI
Monteleone et al.	[21]	Italy	2016	A total of 7 underweight AN patients (1 male), 7 weight-restored AN patients (2 males), and 7 healthy controls (2 males); age: 18–35 yrs	Prospective trial	Enzyme immunoassay
Suchankova et al.	[22]	Sweden and the USA	2015	A total of 84 nondependent drinkers (37 males); race: 72 Caucasian and 12 African American; age: 21–44 yrs	CS	Exposure to a breath alcohol concentration test and brain fMRI during the a Monetary Incentive Delay task
Leggio et al.	[23]	The USA	2014	A total of 45 heavy alcohol drinkers (9 males) randomized to ghrelin or a placebo; race: 14 Black, 24 White, 2 Latino, and 5 Other; age: 25–62 yrs	RCT	Cue-reactivity procedure and AVAS
Scholtz et al.	[24]	Ireland and the UK	2014	A total of 21 RYGB (4 males), 20 BAND (1 male), and 20 BMI-matched unoperated controls (3 males); race: 41 European Caucasian; age: 20–59 yrs	CS	Brain structural MRI and fMRI during exposure to food pictures, radioimmunoassays, ELISA, and mass spectroscopy
Sun et al.	[25]	The USA and Germany	2014	A total of 32 healthy subjects (14 males); age: 18–39 yrs	CS	Brain fMRI after food delivery and radioimmunoassays
van Bloemendaal et al.	[26]	The Netherlands	2014	A total of 16 obese T2DM subjects (8 males), 16 normoglycemic obese subjects (8 males), and 16 healthy lean individuals (8 males); race: all Caucasian; age: 40–70 yrs	RCT	fMRI

Abbreviations: AUD, alcohol-use disorder; AN, anorexia nervosa; FA, food addiction; LSG, laparoscopic sleeve gastrectomy; MDD, major depressive disorder; ADHD, attention deficit hyperactivity disorder; RYGB, Roux-en-Y gastric bypass; BAND, gastric banding; BMI, body mass index; T2DM, type 2 diabetes mellitus; CS, cross-sectional; RCT, randomized controlled trial; ELISA, enzyme-linked immunosorbent assay; fMRI, functional magnetic resonance imaging; AVAS, alcohol visual analogue scale; RIA, radioimmunoassay.

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
