# Peer review of "The Contribution of the Brain–Gut Axis to the Human Reward System"

_biomedicines, 2024, doi:10.3390/biomedicines12081861_

Round 1

Reviewer 1 Report

Comments and Suggestions for Authors

Comments on the Quality of English Language

Please include information about the study's novelty and significance. Some sentences are too long and should be split into two or more. Overall, the review is well written and can be modified and resubmitted after modification.Please include information about the study's novelty and significance. Some sentences are too long and should be split into two or more. Overall, the review is well written and can be modified and resubmitted after modification.

Author Response

Δείτε το συνημμένο 

Reviewer 2 Report

Comments and Suggestions for Authors

The brain-gut axis, a complex network involving the gut microbiota, enteric nervous system, and the central nervous system, is crucial for maintaining energy balance and health. The manuscript reviews the literature from the past decade on the contribution of the brain-gut axis to the human reward system. The results provide valuable insights into the pathophysiology of reward processing impairments, specifically focusing on the roles of enteric neuropeptides and intestinal microbiota. The study holds significant potential for advancing our understanding of how the brain-gut axis influences reward processing. The research is very meaningful, but I have a few questions and suggestions.

1.       The author only included papers from Europe and North America, excluding those from Asia and Africa.

2.       The study population lacks ethnic diversity.

3.       The information in Table 2 is too simplistic.

4.       The discussion section is overly brief, and there are many data points in Table 2 that are worth exploring further. For example, age.

Comments on the Quality of English Language

 Minor editing of English language required

Author Response

Δείτε το συνημμένο 

Reviewer 3 Report

Comments and Suggestions for Authors

The review focuses on the contribution of the brain-gut axis to the human reward system. The aim of the current review is to identify and critically appraise original research on the relationship between the brain-gut axis and the human reward system. This topic is very relevant today. The authors argue that this review will facilitate further study of the pathophysiological links between the brain-gut axis and the reward network structure and function may provide new therapeutic targets in various medical and psychiatric diseases in order to more effectively treat patients' symptoms and alleviate their discomfort.

1. Section 2 should be removed as it duplicates the data presented in Table 2.

2. If the authors provide data on how exactly the selection of articles was made, then it is necessary to provide information on which databases/search engines were used.

3. For the statement (lines 32-35), references from literature should be added.

4. It is necessary to explain why in the section "Psychiatric disorders" only alcohol dependence, ADHD and depression were considered. Why not other types of mental disorders or neurodegenerative diseases?

Comments on the Quality of English Language

Minor editing of English language required.

Round 2

Reviewer 3 Report

Comments and Suggestions for Authors

I thank the authors for their attentive attitude to the comments. As I wrote earlier, it is necessary to justify why the authors paid attention only to the psychiatric disorders they cited. Add information at the beginning of the section that as a result of the literature analysis, only data on the described psychiatric disorders were found in light of the review topic.
